# A Phenomenological Model for Creep and Creep-Fatigue Crack Growth Rate Behavior in Ferritic Steels

Ashok Saxena

Department of Mechanical Engineering, University of Arkansas, Fayetteville, AR 72701, USA; asaxena@uark.edu

**Abstract:** A model to rationalize the effects of test temperature and microstructural variables on the creep crack growth (CCG) and creep-fatigue crack growth (CFCG) rates in ferritic steels is described. The model predicts that as the average spacing between grain boundary particles that initiate creep cavities decrease, the CCG and CFCG rates increase. Further, the CCG data at several temperatures collapse into a single trend when a temperature compensated CCG rate derived from the model is used. The CCG and CFCG behavior measured at different temperatures is used to assess the effects of variables such as the differences between the base metal (BM), weld metal (WM), and heat-affected zone (HAZ) regions. The model is demonstrated for Grade 22 and Grade 91 steels using data from literature. It is shown that differences between the CCG behavior of the Grade 22 steel in new and ex-service conditions are negligible in the BM and WM regions but not in the HAZ region. The CCG behavior of the Grade 91 steels can be separated into creep-ductile and creep-brittle regions. The creep-brittle tendency is linked to the presence of excess trace element concentrations in the material chemistry. Significant differences found in the CCG rates between the BM, WM, and HAZ regions of the Grade 91 steel are explained.

**Keywords:** creep; fatigue; Grade 22 steel; Grade 91 steel; weldments; crack growth; cavitation

## 1. Introduction

Many components of power generation systems such as seam-welded and seamless piping, steam outlet headers, large casings, and rotor forgings operate at high temperatures where creep is a significant design consideration. Several of these components are fabricated from Grade 22 (2.25Cr-1Mo) and Grade 91 (9Cr-1Mo-V-Nb-N) ferritic steels. The rates at which cracks in these structural components are expected to grow under high temperature creep and creep-fatigue conditions are a critical aspect of fitness for service (FFS) evaluations, as recommended in API 579/ASME FFS-1 [1] and BS 7910 [2]. Thus, crack growth rates are an essential input for a temporal understanding of when an inspection of the component is warranted, and after inspection, if the component should be allowed to run, repaired, or retired.

The creep crack growth (CCG) rate behavior and the creep-fatigue crack growth (CFCG) rate behavior previously published in the literature on the above two ferritic materials are critically examined in this paper using a new approach. It is important to understand how the CCG and CFCG rates change with (a) the service temperature and the exposure to high temperatures over a long time during the service, (b) the variations in chemistry such as trace elements and impurity content, and (c) the microstructural gradients that are present in weldments. The proposed model facilitates a closer examination of the influence of these factors on the CCG and CFCG behavior of these materials.

The creep and creep-fatigue crack growth rate tests on the Grade 22 and Grade 91 steels are often conducted over a wide range of temperatures where the CCG rates are correlated with $C^*$ [3–5] or $C_t$ [6], and the CFCG rates are correlated with $(C_t)_{avg}$ [7]. Moreover, $C^*$ is valid for characterizing the CCG rates under widespread creep conditions only [3,4], while $C_t$ is valid under small-scale conditions and becomes identical to $C^*$ under widespread

creep conditions [6]. Thus, $C_t$ is a more general parameter and C* can be considered its subset. Researchers have been reporting the CCG rate, da/dt, correlated with C* without considering whether the specimen was under small-scale-creep or under -widespread creep. Fortuitously, the expressions for estimating $C_t$ and C* for the primarily used compact type specimens [8], using the applied load and the measured load-line displacement rates, are found to be very similar [6]. Therefore, the crack growth rate, da/dt, correlated with $C_t$ or C* as in Equation (1) published in the literature, using compact type specimens, have maximum errors in the values of C* or $C_t$ of approximately 15%. Thus, when using literature data, no distinction is made between the pooled CCG rate data that have been correlated with C* or with $C_t$.

$$\frac{da}{dt} = c(C_t, \ C^*)^q \tag{1}$$

For compact type specimens, C* and $C_t$ under small-scale-creep, $(C_t)_{ssc}$, are given by Equations (2) and (3) below [5,8]:

$$C^* = \frac{P\dot{V}_c}{B(W-a)} \frac{n}{n+1} (2 + 0.522(1 - a/W)) \tag{2}$$

$$(C_t)_{ssc} = \frac{P\dot{V}_c}{B(W-a)} \left(1 - \frac{a}{W}\right) \frac{F'}{F} \tag{3}$$

where $F = (K/P)BW^{1/2}$, $F' = dF/d(a/W)$

$$\dot{\varepsilon}_{ss} = A\sigma^n \tag{4}$$

In the above equations, the various symbols are as follows:

c = an empirical constant in the relationship between the time-rate of crack growth, da/dt, and C* or $C_t$, see Equation (1),

q = the exponent in the above relationship also empirically determined,

P = the applied load,

$V_c$ = the load-line displacement due to the creep and the dot indicates the rate of change with time,

A = the crack size,

B = the specimen thickness,

W = the width of the specimen,

σ = the applied stress,

$\dot{\varepsilon}_{ss}$ = the secondary creep rate,

n = the exponent in the power-law empirical relationship, see Equation (4), between applied stress and the secondary creep rate,

A = the pre-exponent constant in the above relationship, and

K = the stress intensity parameter.

It is noted that the value of q is between 0.5 and 0.9, which further suppresses the error associated with using C* and $C_t$ without the consideration of the scale of creep deformation in compact type, C(T), specimens. Under creep-fatigue loading conditions, the average value of $C_t$ during the hold time of a trapezoidal wave form, $(C_t)_{avg}$, is used to correlate the CFCG rates [7].

## 2. Phenomenological Model for Creep Crack Growth

The model presented here evaluates the effects of the variables that are expected to influence the CCG behavior in ferritic steels. This model was first proposed by Wilkinson and Vitek [9], developed further by Saxena and Bassani [10], and is once again refined in this paper. The model is referred to as the WVSB model. In the model, it is assumed that the steady-state creep deformation conditions given in Equation (4) prevail in the cracked body.

Figure 1a schematically shows the idealized development of creep damage ahead of the crack tip in the form of an array of creep cavities with radii of $\rho_1$, $\rho_2$ ....$\rho_i$ ... $\rho_N$ that

are spaced by a center-to-center distance of 2b. The cavities are assumed to nucleate on the grain boundary facets, see Figure 1b, which are aligned normal to the loading direction and have an initial diameter of $\rho_N$, corresponding to the radius of the nucleating particle. These cavities grow at a rate that is constrained by the power-law creep, see Equation (4), in the crack tip stress environment. When the cavity closest to the crack tip approaches a critical radius, it coalesces with the crack tip and the crack is believed to have advanced by 2b. All successive cavities grow and move closer to the crack tip, while the one nearest becomes part of the crack. A steady-state crack growth rate is established and described by Equation (5) [9,10]:

$$\frac{da}{dt} = \frac{(2b)^{\frac{2n+3}{n+1}} 3d.A^{\frac{1}{n+1}}}{2.5\left(\rho_c^3 - \rho_N^3\right)\sum_{m=1}^{m_1}(m)^{\frac{n}{n+1}}}\left(\frac{C^*}{I_n}\hat{\sigma}_{yy}(90^0,\ n)\right)^{\frac{n}{n+1}} \tag{5}$$

where
  2b = the inter-cavity spacing,
  m = the number of cavities in the process zone ahead of the crack tip that grow due to creep deformation (approximately between 3 to 5),
  $\rho_i$ = the radius of the $i^{th}$ cavity from the crack tip,
  $\rho_c$ = the critical cavity radius,
  $\rho_N$ = the cavity radius at nucleation,
  d = the grain diameter, and
  $\hat{\sigma}_{yy}(90^0,\ n)$ = the crack tip field quantity that depends on n, is evaluated at an angle of 90 degrees to the crack plane and has a value on the order of 1.

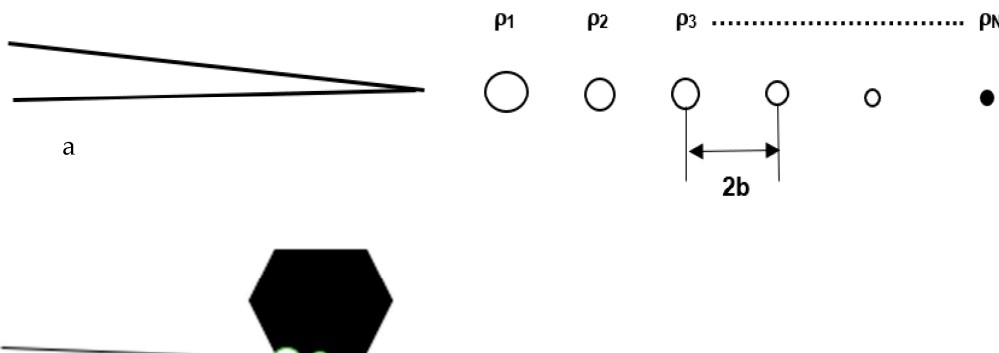

**Figure 1.** (**a**) Array of cavities ahead of a growing creep crack and (**b**) creep cavities located on grain boundary facets normal to the loading axis.

$$I_n = 6.568 - 0.4744n + 0.0404n^2 - 0.00262n^3$$

If it is assumed that for creep-ductile materials, $\rho_c \gg \rho_N$ and $\rho_c \approx b$. Equation (5) then reduces to:

$$\frac{da}{dt} = \beta\alpha(n)(A)^{\frac{1}{n+1}}\left(\frac{C^*}{b}\right)^{\frac{n}{n+1}} \tag{6}$$

where $\alpha(n) = \dfrac{(2)^{\frac{2n+3}{n+1}}(1.2)}{I_n^{\frac{n}{n+1}}}$ and $\beta = \dfrac{d\left(\hat{\sigma}_{yy}(90^0,\ n)\right)^{\frac{n}{n+1}}}{\sum_{m=1}^{m_1}(m)^{\frac{n}{n+1}}}$.

The constant $\beta$, as seen above, also includes the angular terms in the crack tip stress fields and can be made part of a consolidated constant. Moreover, $\alpha(n) \approx 13$ for $5 \le n \le 10$ and the term $\sum_{m=1}^{m_1}(m)^{\frac{n}{n+1}} \approx 13.0$ for $5 \le m_1 \le 10$ can also be merged into the same con-

solidated constant. If $b_0$ represents the inter-cavity spacing at which da/dt asymptotically approaches a lower bound value where $\rho_c \approx b_0$, it implies a highly creep-ductile material. Equation (3) can be written as:

$$\frac{da}{dt} = c_1(A)^{\frac{1}{n+1}}(b_0/b)^{\frac{n}{n+1}}(C^*)^{\frac{n}{n+1}} \tag{7}$$

where $c_1 = \dfrac{\alpha(n)\beta}{(b_0)^{\frac{n}{n+1}}}$.

In Equation (7), the term $(A)^{\frac{1}{n+1}}$ compensates the CCG rate behavior for temperature. For b less than $b_0$, the CCG rates are expected to be higher. Thus, a microstructural length dimension is explicitly included in the CCG rate equation. This microstructural parameter could also evolve during the service and account for changes in CCG properties due to exposure to high temperatures during the service. For example, if during the service, new grain boundary particles form due to exposure to the service temperatures and reduce the value of b, the CCG rate is expected to increase, and the material is expected to embrittle during the service.

A ductility exhaustion model has been proposed by Nikbin, Smith, and Webster (NSW) [11] that relates creep ductility to the CCG behavior as shown in Equation (8) below:

$$\frac{da}{dt} = \frac{n+1}{\varepsilon_f^*}A^{\frac{1}{n+1}}\left[\frac{C^*}{I_n}\right]^{\frac{n}{n+1}}r_c^{\frac{1}{n+1}} \tag{8}$$

where $\varepsilon_f^*$ is the multi-axial creep ductility and $r_c$ is the process zone size of the material that depends on the microstructural characteristics and the crack tip constraint. There are similarities between the two models as both equations can be reduced to Equation (9). The WVSB model explicitly contains microstructural terms while the NSW model is based on notional parameters, such as $r_c$.

$$\frac{da}{dt} = c_1(A(T))^{\frac{1}{n+1}}(C^*)^{\frac{n}{n+1}} \tag{9}$$

Constant A is a strong function of temperature. However, when it is raised to a power of 1/(n + 1), where n ranges between 5 and 13 for ferritic steels, the dependence of da/dt on A becomes weak but remains significant. Further, we can replace $C^*$ with a more general parameter $C_t$ in Equation (9), making it also applicable to small-scale-creep conditions and becoming identical to $C^*$ under steady-state creep conditions. This parameter also permits the inclusion of the CFCG rates at various hold times on the same plot [6,7]. We next define a reference temperature, $T_{ref}$, that is equal to, say, a commonly used temperature for the material and for which the CCG data are available. At $T_{ref}$,

$$(da/dt)_{ref} = c_1(A(T_{ref}))^{1/(1+n_{ref})}C_t^q \tag{10}$$

In Equation (10), n/(n + 1) is replaced by an empirically determined material constant, q. The value of n/(n + 1) ranges from 0.75 to 0.9 for n values between 4 and 10 and compares well with the range of values of q found in the CCG data for a wide variety of steels.

Next, we divide Equation (9) by Equation (10) to get:

$$\omega(T) = \frac{da/dt}{(da/dt)_{ref}} = \frac{(A(T))^{(1/(1+n))}}{(A(T_{ref}))^{(1/(1+n_{ref}))}} \tag{11}$$

and

$$da/dt = \omega(da/dt)_{ref} \tag{12}$$

We can then write:

$$\frac{da^*}{dt} = \frac{1}{\omega}\frac{da}{dt} = c_1(A(T_{ref}))^{(1/(1+n_{ref}))}C_t^q \tag{13}$$

The parameter $\omega(T)$, a function of temperature, is referred to as the temperature compensation parameter and is applied to the CCG and CFCG data at a variety of temperatures, so the effects of temperature can be separated from the effects of other variables on the CCG and CFCG rates. Thus, the CCG equation can be written as:

$$(da/dt)_{ref} = da/dt* = \frac{1}{\omega}\frac{da}{dt} = c_1(A(T_{ref}))^{(1/(1+n_{ref}))}C_t^q = c(C_t)^q \tag{14}$$

where $c = c_1(A(T_{ref}))^{(1/(1+n_{ref}))}$.

The values of c and q are empirically determined constants from the CCG and CFGG data that correspond to the reference temperature. To determine c, the CCG data from several temperatures can be expressed as $da/dt*$ (by dividing $da/dt$ with $\omega$ as in Equation (13)) and $C_t$ and pooled together for the purposes of the regression analysis leading to the c and q values. Results from the analyses are shown for Grade 22 and Grade 91 steels in the next section.

### 3. CCG in Grade 22 and Grade 91 Materials

#### 3.1. CCG Behavior of Grade 22 Materials

The CCG and CFCG data were collected from the literature for the Grade 22 materials [12–17]. Crack growth data were identified separately for the new and ex-service base metals (BM), the new and ex-service weld metals (WM), and for the new and ex-service heat-affected zone regions (HAZ) in the temperature range from 538 °C to 594 °C. These data were plotted using the temperature compensated CCG rates to identify variables other than temperature that are relevant to the CCG rates in this material. The secondary creep constants, A and n, were obtained from the studies that reported the CCG data and are summarized in Table 1. The value of $\omega$, see Equation (11), at each temperature is also reported in Table 1.

**Table 1.** Secondary creep constants for Grade 22 steel at three temperatures [10–16].

| T °C | $\dot{\varepsilon}_{ss}=A(T)\sigma^n$ | | $(A(T))^{1/(1+n)}$ | $\omega(T)$ from Equation (11) |
| --- | --- | --- | --- | --- |
| | **A(T)** | **n** | | |
| 540 (Ref Temp, $T_{ref}$) | $2.2 \times 10^{-20}$ | 6.6 | 0.00259 | 1 |
| 550 | $3.94 \times 10^{-22}$ | 7.79 | 0.00367 | 1.41 |
| 566 | $6.36 \times 10^{-23}$ | 9.36 | 0.007243 | 2.78 |
| 594 | $1.94 \times 10^{-24}$ | 10.08 | 0.00724 | 2.8 |

Figure 2 shows the correlation between the CCG rates at various temperatures for the Grade 22 base metal in the new and ex-service conditions at various temperatures. In Figure 3, the same data are plotted as the temperature compensated crack growth rates referenced to a temperature of 540 °C. All data over several orders of magnitude in growth rates fall within a narrow scatter band in Figure 2 but are even better consolidated in Figure 3. In Figure 2, the crack growth rates at 594 °C are seen to be consistently higher than at other lower temperatures, but when plotted in the temperature compensated plot, they become indistinguishable from the data at other temperatures. No significant differences were found in the data trends in Figure 3 for the new and ex-service BMs. Thus, the mean and upper bound (UB) and lower bound (LB) values of c in Table 2 were determined from the pooled data consisting of the data from all new and ex-service BMs. This allows for direct comparisons of the CCG data in the WM and HAZ regions with the BM. Table 2 lists

the constants c and q in Equation (10) that describe the mean, the upper bound (UB), and the lower bound (LB) trends in the various regions of the weldment.

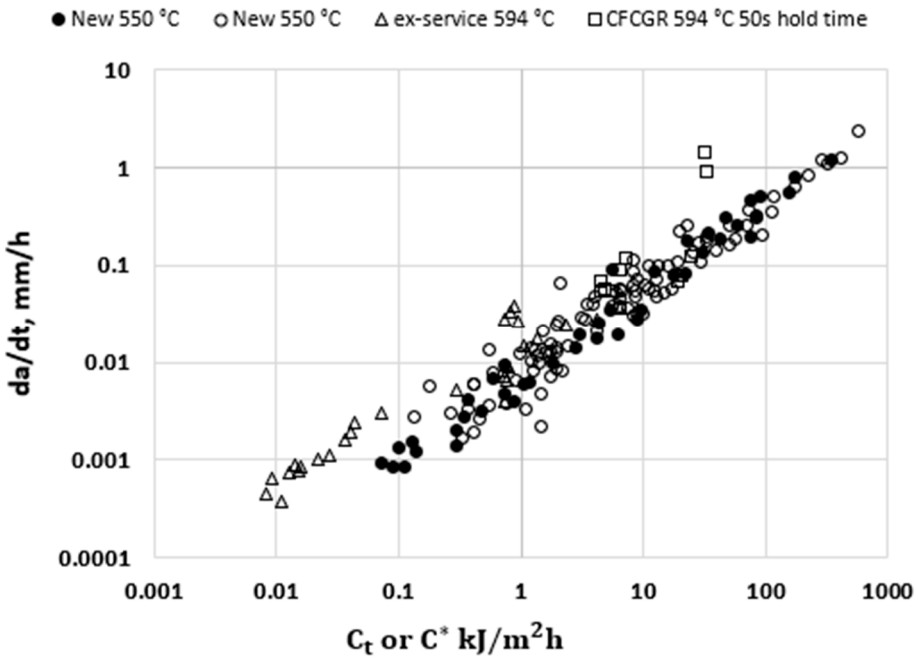

**Figure 2.** Base metal CCGR and CF-CGR behavior as a function of $C_t$ for hold times of 50 s for Grade 22 steels at various temperatures in new and ex-service conditions.

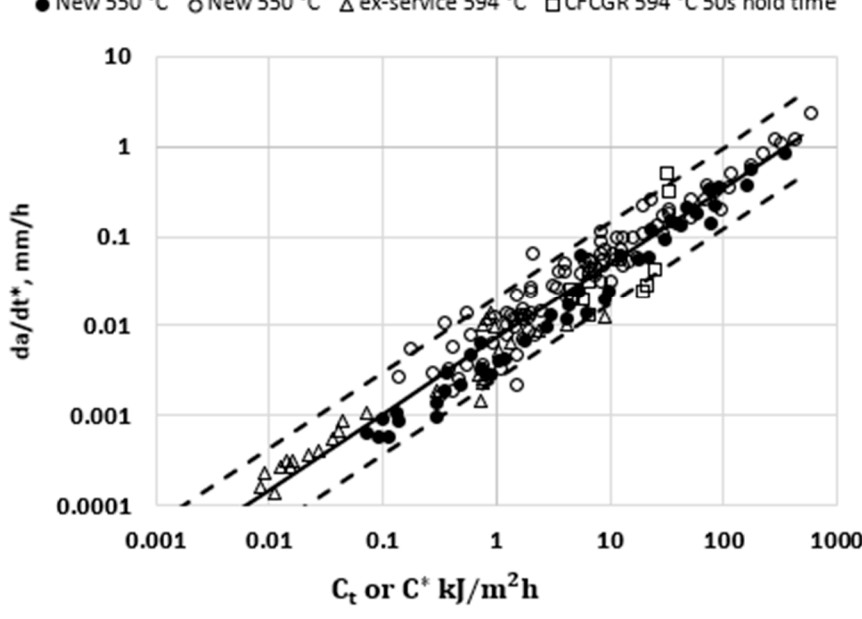

**Figure 3.** Temperature compensated base metal CCG and CFCG behavior at a hold time of 50 s as a function of $C_t$ or $C^*$ for Grade 22 steels at various temperatures are plotted using temperature compensated rates.

The CCG rates for the samples taken from the weld metal (WM) and heat-affected-zone (HAZ) regions of the Grade 22 material are plotted in Figure 4. The data obtained at various temperatures are adjusted for 540 °C using the temperature compensation parameter, $\omega$. For comparison, the 95% confidence interval (CI) for the base material in Figure 3 is also superimposed in Figure 4 and the 95% CI band for just the service exposed HAZ region. The CCG rates in the new and ex-service WMs lie in the center of the BM band, but there

is a clear difference in the CCG rates between the ex-service and new materials when the samples are taken from the heat-affected zone (HAZ) regions. Exposure to the service temperatures appears to degrade the CCG rates in the HAZ regions while apparently not affecting the BM and WM. This is presumably due to the formation of the new carbides or the coarsening of existing carbides due to the long-term exposure to high temperatures that, for a long time, have been related to the embrittlement of these steels. This observation is also consistent with the service experience, where the instance of cracking is most frequently found at the interface of the BM and WM in the seam-welded steam pipes but not in the BM and WM regions, and not in the seamless piping made from the Grade 22 material [17,18].

**Table 2.** CCG and CFCG constants for Grade 22 steels. The values of $C_t/C^*$ are in KJ/m$^2$h and crack growth rates are in mm/h. All constants are referenced to a temperature of 540 °C.

| Material | Temperature °C | Mean/Upper Bound | c | q |
|---|---|---|---|---|
| BM/Ex-service BM | 540–594, referenced to 540 | Mean | 0.0072 | 0.8394 |
| | | UB | 0.0205 | |
| | | LB | 0.0025 | |
| New and ex-service WM, New HAZ | 540, referenced to 540 | Mean | 0.0072 | 0.8394 |
| | | UB | 0.02050 | |
| | | LB | 0.0025 | |
| Ex-service HAZ | 538, assumed to be same as at 540 | Mean | 0.037 | 0.8294 |
| | | UB | 0.00666 | |
| | | LB | 0.0185 | |

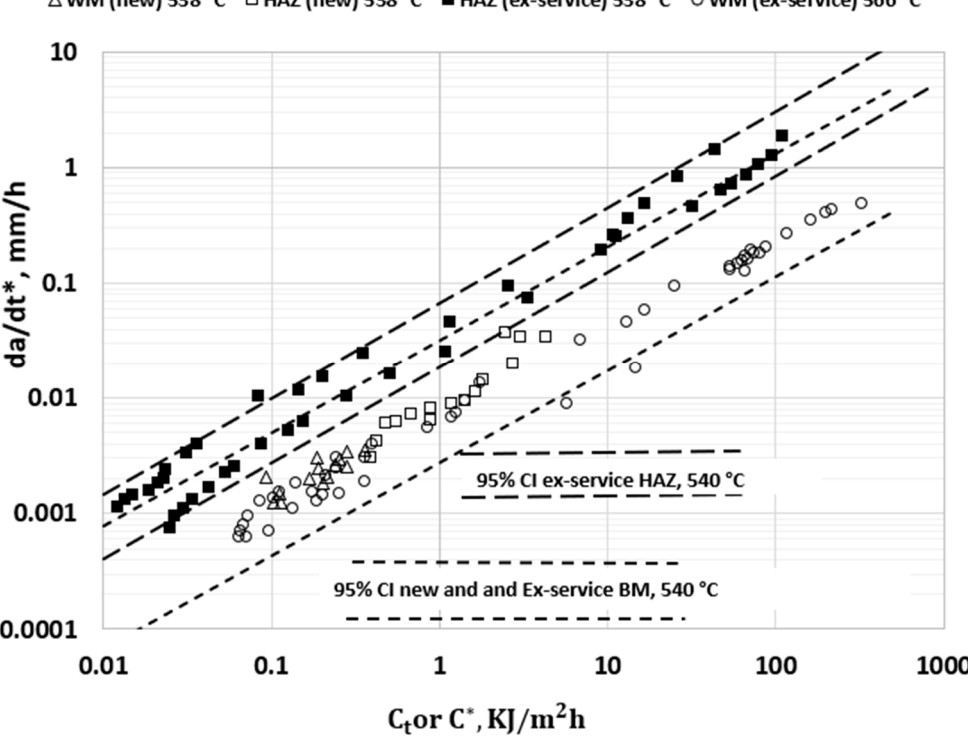

**Figure 4.** CCG behavior of Grade 22 steel in the WM and HAZ regions of weldments in new and ex-service conditions compared to BM. All CCG rates are plotted as temperature compensated values using a reference temperature of 540 °C.

In summary, there are no distinctions in the temperature compensated CCG behavior between the new BM, the ex-service BM, the new WM, the ex-service WM, and the new HAZ regions for the Grade 22 material. However, the CCG rates in the ex-service HAZ regions were found to be higher than the new HAZ region. The constants representing the CCG kinetics are listed in Table 2 for all regions of the Grade 22 weldments.

### 3.2. CCG Behavior of Grade 91 Materials

The CCG behavior of the BM, WM, and HAZ regions of the Grade 91 steels have been reported in references [19–24] in the temperature range of 538 °C to 650 °C. The data at 538 °C, 593 °C [20], 600 °C [21,23], and 625 °C [21] on the BM were available in an excel format. The data at 650 °C and some at 600 °C [22–24] were available as regression constants for the CCG data. The latter were included in the CCG plots as data points estimated from equations at the $C_t/C^*$ values of 0.01, 0.1, 1, 10, and 50 KJ/m$^2$h to document the reported trend, but are not used in the regression analyses for determining the CCG constants because they are not measured values.

Table 3 lists the steady-state or secondary creep behavior for the Grade 91 material at various temperatures ranging from 538 °C to 650 °C, covering the range of temperatures for which the CCG data were available in the literature. The table also provides the computed $\omega(T)$ values for the various temperatures.

**Table 3.** Measured creep deformation behavior at various temperatures and the estimated values of the temperature compensation parameter for CCGR behavior of Grade 91 steel.

| Temperature (°C) | $\dot{\varepsilon}_{ss} = A(T)\sigma^n$ | | $(A(T))^{1/(1+n)}$ | $\omega(T)$ |
| :---: | :---: | :---: | :---: | :---: |
| | **A(T)** | **n** | | |
| 538 (Ref Temp) | $4 \times 10^{-47}$ | 17.536 | 0.00313 | 1.00 |
| 550 (BM) | $1.29 \times 10^{-37}$ | 13.7 | 0.00309 | 0.989 |
| 550 (WM) | $1.07 \times 10^{-31}$ | 11.4 | 0.00318 | 1.016 |
| 550 (HAZ) | $2.45 \times 10^{-35}$ | 12.9 | 0.003236 | 1.034 |
| 575 | $6 \times 10^{-31}$ | 11.202 | 0.003336 | 1.066 |
| 593 | $1 \times 10^{-36}$ | 14.195 | 0.004274 | 1.365 |
| 600 | $1 \times 10^{-26}$ | 9.834 | 0.003982 | 1.272 |
| 625 | $1 \times 10^{-22}$ | 8.383 | 0.004522 | 1.445 |
| 650 (BM) | $1.092 \times 10^{-20}$ | 8.462 | 0.007706 | 2.46 |
| 650 (WM) | $1.37 \times 10^{-20}$ | 7.65 | 0.05054 | 1.615 |
| 650 (HAZ) | $2.3 \times 10^{-20}$ | 8.462 | 0.008404 | 2.685 |

The temperature compensated CCG rates are shown in shown in Figure 5 for the base metal regions of the Grade 91 steel. At the temperatures of 538 °C, 594 °C, 600 °C, and 650 °C, the CCG tests were conducted using new plate material while the tests conducted at 625 °C were performed on a material taken from an ex-service pipe; however, the pipe section used for testing was heat treated prior to testing to rejuvenate its microstructure to the original state [25–27]. The latter should, therefore, also be considered a new material. Large variability in the CCG rates were observed between the various data sets with the rates being much higher for some of the 600 °C from Yatomi et al. [20] and all the 625 °C tests from Shingledecker [21] compared to the other conditions reported. No systematic variations were observed in the CCG rates with temperature since the crack growth rates plotted are compensated for temperature. At 600 °C, the data reported by Yatomi et al. [20] showed much higher CCG rates than the data reported by Kim et al. [23]. Such large variability cannot simply be attributed to random experimental scatter and therefore deserves a more in-depth discussion.

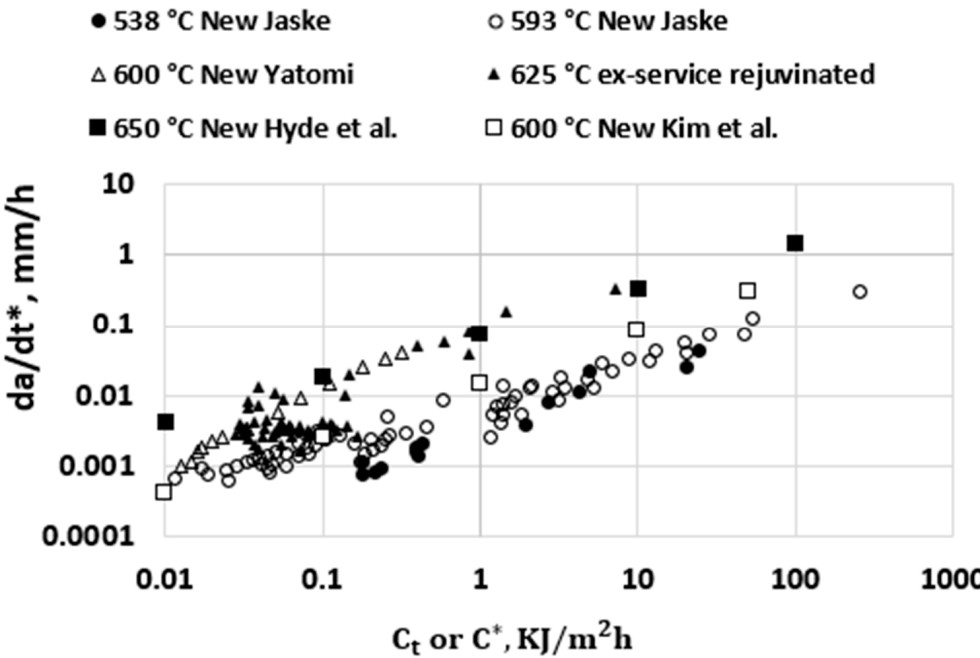

**Figure 5.** Temperature compensated CCG rates in Grade 91 BM in new and service exposed but rejuvenated conditions.

Parker and Siefert [28] have conducted an exhaustive study on the levels of certain trace elements that include S, Cu, Sn, As, Sb, and Pb present in the Grade 91 steels as impurities and their effect on the creep behavior, specifically the creep ductility. They have reported a tendency of components made from heats containing more than critical levels of the trace elements to develop early cracking during the elevated temperature service. They separated heats of the Grade 91 material between what they termed as creep damage resistant and creep damage prone and related them to chemistry, particularly the amounts of the above trace elements.

Table 4a lists the chemical composition of the heat of the Grade 91 steel that experienced early cracking during the service used by Shingledecker [21] in his CCG studies in the service's exposed but rejuvenated condition. Thus, after the removal from the service, the material was heat treated to rejuvenate its microstructure to the original form prior to machining the CCG samples that were tested at 625 °C [21,25–27]. Table 4b lists the actual levels of the trace elements S, Cu, Sn, As, Sb, and Pb along with levels that were seen to result in the creep damage resistant and creep damage prone behaviors in the Parker and Seifert [28] study; in other words, creep-ductile or creep-brittle behavior, respectively. The compositions shown in Table 4b clearly show that the levels of several trace elements in the heat of Grade 91 that exhibited creep-brittle behavior did exceed or had comparable levels of trace elements known to cause creep-brittle behavior. This is consistent with the experimental observations.

In Figure 6, all tests yielding the creep-ductile trend in Figure 5 were plotted using one type of marker, and tests yielding creep-brittle trends were plotted using a different marker. The difference between the two trends is obvious in the plot. Separate 95% CI bands were also developed for each of the two conditions as shown in Figure 6. The creep-brittle behavior also leads to higher scatter in the CCG data compared to the creep-ductile behavior. The CCG constants for the creep-ductile and creep-brittle conditions are listed in Table 5 for the Grade 91 steels.

**Table 4.** a: Actual and nominal chemical composition of test material (in weight%): NR = Not Reported. b: Actual trace element content in the material used by Shingledecker [22] compared to levels that are known to cause creep damage resistant and creep damage prone conditions in Grade 91 steels [28].

**(a)**

| | | | | | | | Chemical Composition (wt %) | | | | | | | | | |
|---|---|---|---|---|---|---|---|---|---|---|---|---|---|---|---|---|
| | c | Si | Mn | P | S | Ni | Cr | Mo | As | V | Nb | Al | Cu | N | Sb,Sn | Fe |
| Shingledecker | 0.11 | 0.31 | 0.45 | 0.011 | 0.009 | 0.19 | 8.22 | 0.94 | 0.005 | 0.21 | 0.07 | 0.006 | 0.16 | 0.039 | 0.001 | Bal |
| Hyde et al. BM | 0.11 | 0.022 | 0.36 | NR | NR | NR | 8.74 | 0.98 | NR | 0.21 | 0.12 | NR | 0.08 | 0.048 | NR | Bal |
| Hyde et al. WM | 0.087 | 0.28 | 1.04 | NR | NR | NR | 8.6 | 1.02 | NR | 0.22 | 0.24 | NR | 0.03 | 0.04 | NR | Bal |
| Kim et al. BM | 0.115 | 0.23 | 0.415 | 0.012 | 0.014 | 0.22 | 8.9 | 0.87 | NR | 0.0194 | 0.073 | 0.02 | 0.038 | 0.0513 | NR | Bal |
| Nominal | 0.1 | 0.38 | 0.46 | 0.02 | 0.002 | 0.33 | 8.1 | 0.92 | - | 0.18 | 0.073 | 0.034 | - | 0.049 | - | Bal |

**(b)**

| | | | | Chemical Composition (wt %) | | |
|---|---|---|---|---|---|---|
| | S | Cu | Sn | As | Sb | Pb |
| Actual | 0.009 | 0.16 | 0.001 | 0.005 | 0.001 | ? |
| Damage Prone | 0.01 | 0.19 | 0.008 | 0.0128 | 0.0023 | 0.00075 |
| Damage Resistant | 0.002 | 0.05 | 0.003 | 0.0042 | 0.00063 | 0.00003 |

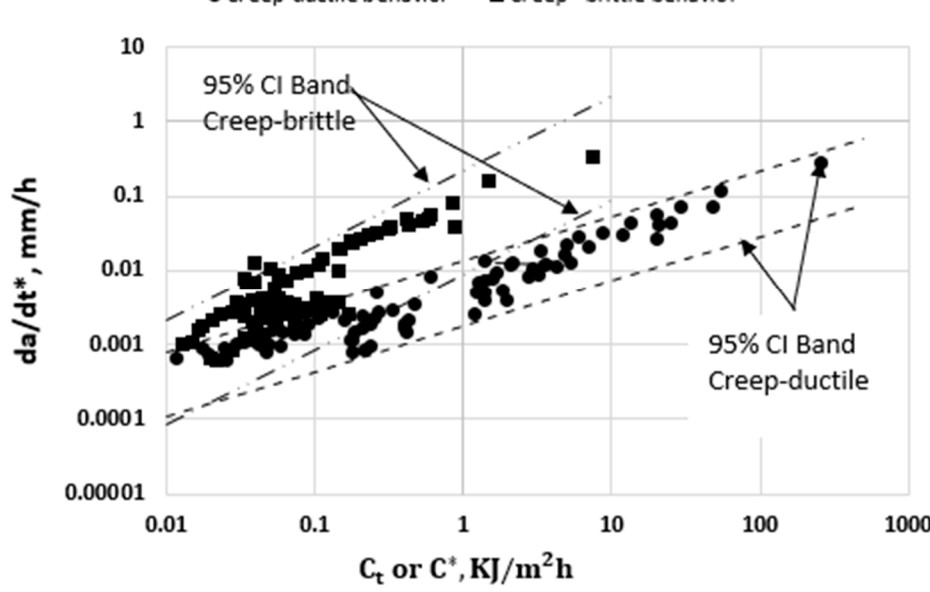

**Figure 6.** The 95% confidence interval bands for temperature compensated CCG rates in Grade 91 BM in the temperature range from 538 °C to 650 °C for conditions exhibiting creep-ductile and creep-brittle behavior. All CCG rates are referenced to 538 °C.

In Figure 7, the additional CCG data at the temperatures of 550 °C, 600 °C, and 650 °C from the literature on [22–24] on the Grade 91 BM are plotted. The markers in this figure are values derived from the equations representing the mean trends in the various sets of data. All the CCG rates appear to fall within the 95% CI band for the creep-ductile materials in Figure 6, with the 650 °C CCG data lying near the upper scatter band for the creep-ductile materials. The chemistry of the base metals used for these CCG tests conducted at 650 °C by Hyde et al. [22] and at 600 °C and 550 °C conducted by Kim et al. [23,24] are provided in Table 4. However, the chemical analysis for the trace elements was not reported in either of these studies. Thus, judgements about whether creep-ductile or creep-brittle behavior is expected in these experiments cannot be made. Based on just the CCG data trend, these

materials are classified as creep-ductile. Next, the CCG trends in the WM and HAZ regions of the weldments that use the above materials as the BM are discussed.

**Table 5.** CCG rate constants for creep-ductile and creep-brittle behaviors in Grade 91 steels. All constants are for a reference temperature of 538 °C for $C_t$ expressed in KJ/m$^2$h and da/dt* in mm/h.

| Material | Temperature °C | Mean/Upper Bound | c | q |
|---|---|---|---|---|
| CCG-Creep-ductile behavior | 538–593, referenced to 538 | Mean | 0.0048 | 0.6093 |
| | | UB | 0.01314 | |
| | | LB | 0.001752 | |
| CCG-Creep-brittle behavior | 600–625, referenced to 538 | Mean | 0.0427 | 0.9991 |
| | | UB | 0.2130 | |
| | | LB | 0.008557 | |
| CFCG | 625 but referenced to 538 | Mean | 0.0186 | 0.5061 |
| | | UB | 0.1099 | |
| | | LB | 0.003149 | |

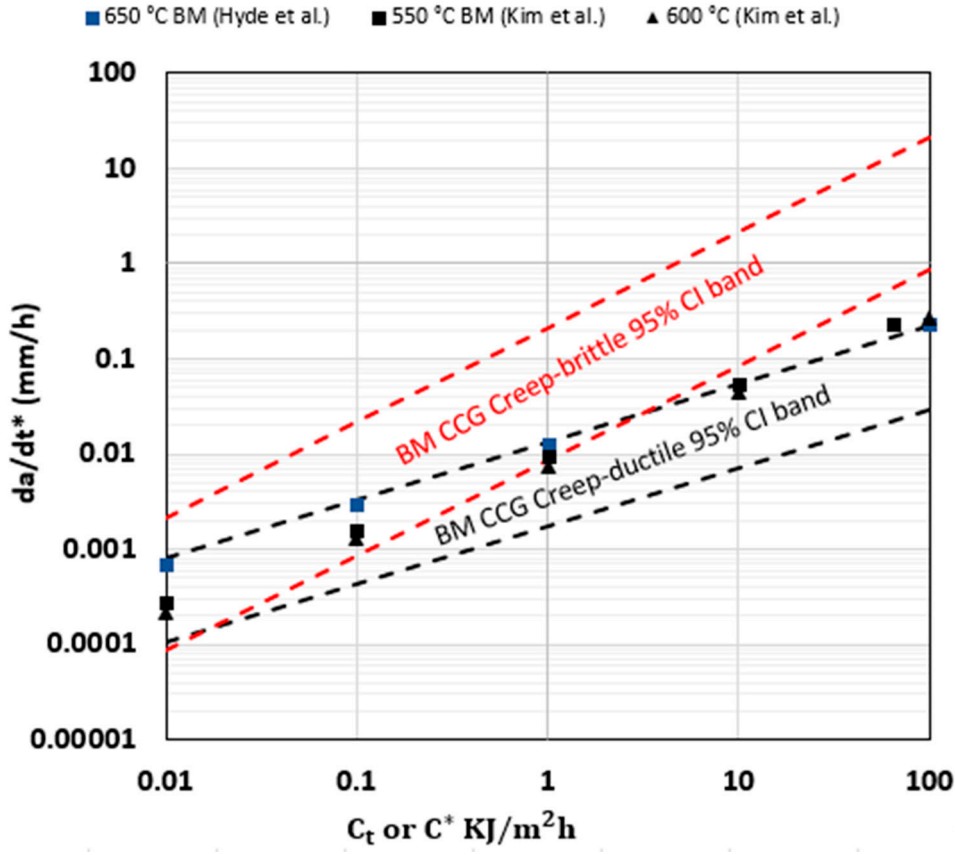

**Figure 7.** Additional Grade 91 BM data in the temperature range of 550 °C, 600 °C, and 650 °C [22–24] compared to the 95% CI bands for creep-ductile and creep-brittle shown in Figure 6.

Figure 8 shows the CCG trends in the WM and HAZ regions of welds in Grade 91 steel in the temperature range of 550 °C to 650 °C [22,23]. The WM and HAZ regions have nearly identical CCG behaviors at 550 °C and 600 °C [23,24] but the rates were somewhat higher when compared to the BM at those temperatures. At 650 °C, the CCG rates taken from a different study [22] in the HAZ region of the weldment show much higher crack growth rates compared to the BM at that temperature. The CCG rates in the HAZ region in

this latter study [22] are in the creep-brittle regime, and significantly higher than at 550 °C and 600 °C data for WM, HAZ, and BM. However, it cannot be concluded that it is the temperature that causes the creep-brittle behavior because these data were obtained on a different P91 Grade steel. The WM and BM chemistry of the weldments are shown in Table 4, but the amounts of trace elements are missing. Therefore, the higher CCG rates at 650 °C in the HAZ region cannot also be attributed to higher content of trace elements in the composition of the weld or base metal and the cause remains an unanswered question.

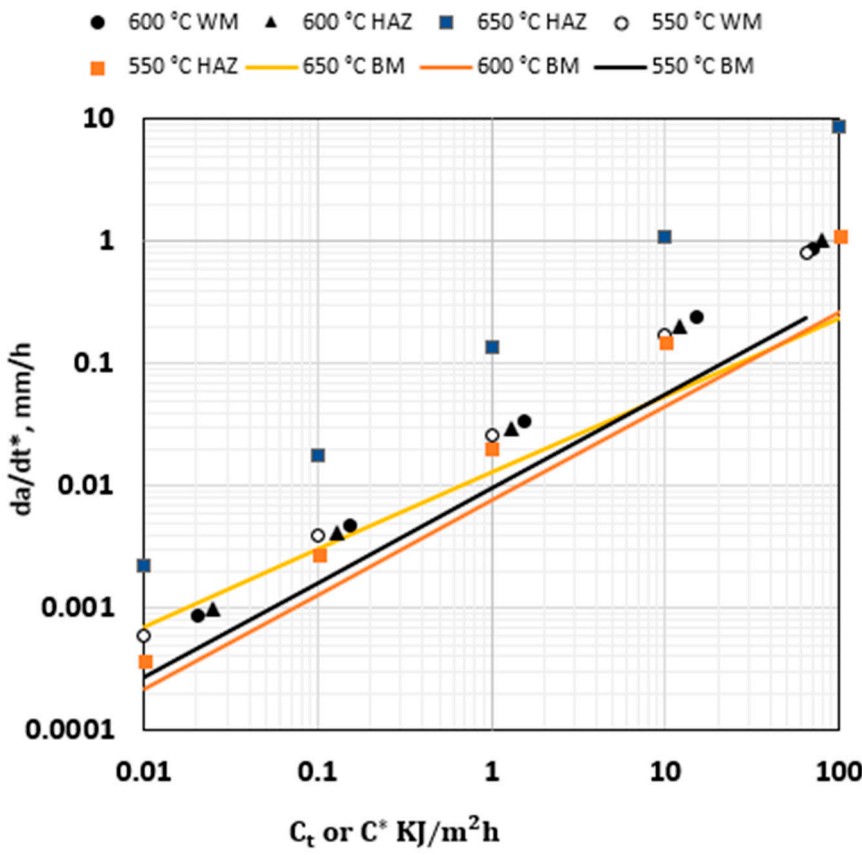

**Figure 8.** CCG behavior of the WM and HAZ regions of Grade 91 steel welds in the temperature range of 550 °C to 650 °C are compared to the CCG of the base metal used in fabricating the weldments.

## 4. CFCG in Grade 22 and Grade 91 Steels

### 4.1. CFCG in Grade 22 Steel

In Figures 2 and 3, the CFCG data available for the Grade 22 material for a hold time of 50 s at a temperature of 594 °C are also included [12]. The time rate of the crack growth, $(da/dt)_{avg}$, during the hold time was correlated with the average value of the $C_t$ parameter, referred to as $(C_t)_{avg}$. The reported CFCG rates are also temperature compensated in Figure 3 such as the CCG data. The $(da/dt)_{avg}$ and $(C_t)_{avg}$ data were primarily obtained under small-scale-creep (SSC) conditions, but are seen to blend with the CCG data, especially in Figure 3 after temperature compensation. Thus, no separate correlation for the CFCG is needed for this material. The cyclic crack growth rate for a hold time $t_h$ can simply be estimated by multiplying the $(da/dt)$ value with $t_h$. This observation is significant for evaluating the lives of components that are periodically cycled during a service such as high-pressure equipment.

### 4.2. CFCG in Grade 91 Steel

There is large scatter in the CFCG data shown in Figure 9 [25–27], like that observed in the CCG rates on the same heat of the Grade 91 steel in the rejuvenated ex-service condition. Some of this scatter may be attributed to tests being conducted by 13 laboratories

around the world with different levels of prior experience in conducting CFCG testing, but a significant portion must also be from the material itself. All the CCG data shown in Figure 5 at 625 °C show considerable scatter even when all tests were conducted by a single and highly experienced laboratory. The 60 s hold time data seem to show somewhat lower CFCG rates compared to the 600 s hold time. However, these differences are small in comparison to the general scatter in the data. For the regression analyses to obtain the mean and 95% confidence intervals, the data from both hold times were pooled. The constants representing the mean and the upper and lower bounds of the 95% confidence intervals for the CFCG behavior are listed in Table 3. The reason for the large scatter in the CCG rates in the Grade 91 steel at 625 °C was discussed previously as embrittlement caused by higher concentrations of trace elements in the chemistry of the steel. The same factors are expected to contribute to the scatter in the CFCG trend.

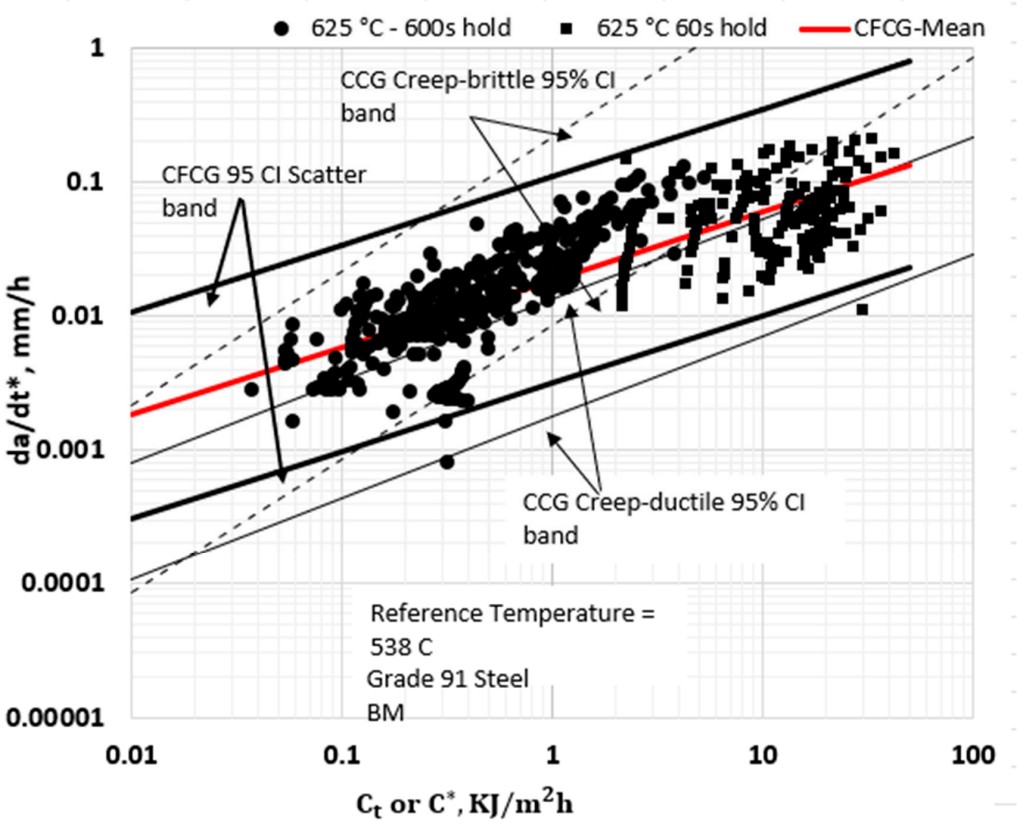

**Figure 9.** Temperature compensated CFCG behavior of Grade 91 steel at 625 °C at hold times of 60 s and 600 s. The 95% CI bands for creep-ductile and creep-brittle CCG behavior are superimposed on the CFCG data for comparison.

The CFCG trend lies between the creep-ductile and creep-brittle CCG trends for the Grade 91 steel. The slope is better aligned with the creep-ductile trend than with the creep-brittle trend and the crack growth rates are closer to the upper bound creep-ductile behavior. It appears that under cyclic loading, the crack growth behavior tends more toward the creep-ductile behavior. The cyclic softening of the material in the crack tip region may be responsible for the tendency toward creep-ductile behavior.

## 5. Summary and Conclusions

The following are the summary and conclusions that can be gleaned for this study:

- A phenomenological model for rationalizing the effects of the temperature and microstructural characteristics on the CCG and CFCG behavior of ferritic steels is proposed in this paper and was evaluated using extensive amounts of data gathered from the literature. The ability of the model to rationalize the effects of the temperatures is

clearly demonstrated. The model contains a characteristic microstructural parameter that can only be determined by a quantitative microstructural evaluation of the test materials, typically not reported in CCG studies.

- The base metal (BM), CCG, and CFCG data for the Grade 22 steels in the new and ex-service materials in the temperature range of 538 to 594 °C were indistinguishable.
- The weld metal (WM) CCG data for the Grade 22 steels in the new and ex-service conditions on average followed the same trend as the BM.
- The CCG rates in the heat-affected zone (HAZ) region of the Grade 22 materials are comparable to the CCG rates in the BM and WM, but the CCG rates after the service exposure were on average approximately four times higher.
- The CFCG and CCG behaviors in the Grade 22 steels were indistinguishable from each other.
- Extensive amounts of the BM, CCG, and CFCG data for the new and rejuvenated ex-service Grade 91 steels in the temperature range of 538 °C to 650 °C are reported from the literature. The tests were conducted on four separate heats of material.
- The variability among the BM and CCG behaviors in the Grade 91 steels was significantly higher than that found in the Grade 22 steels.
- It was observed that the CCG in the BM of the Grade 91 steels at the temperatures of 538 °C and 593 °C, along with some at 600 and at 650 °C, follow creep-ductile tendencies, implying a higher CCG resistance than those under the creep-brittle behavior observed in some of the heats tested at 600 °C and 625 °C. It is hypothesized, based on results from a study conducted by Parker and Seifert [28], that subtle differences in chemical compositions among the trace impurity elements can cause creep-brittle tendencies. This strongly suggests the need for reporting the full chemistry, including levels of trace elements, as a standard practice when reporting the CCG and CFCG test results.
- The CCG constants for the Grade 22 and Grade 91 steels representing the mean, upper bound (UB), and lower bound (LB) trends were calculated and reported in the paper. For Grade 91, the constants were reported separately for the creep-ductile and creep-brittle trends.

## 6. Recommendations for Future Work and Conclusions

The collection of the CCG and CFCG data reported in this study provides a new way to report CCG and CFCG data such that it is compensated for effects due to temperature. The temperature compensation parameter has been derived based on a phenomenological model presented as part of this paper and is evaluated extensively using the data from several prior studies. Some recommendations for future work are listed below.

- Conduct a round-robin study of CCG and CFCG testing consisting of several laboratories, using a well-established cavitation resistant material with adequately documented creep deformation behavior, tensile data, a well characterized microstructure, documented chemistry including levels of trace elements that cause embrittlement, and cyclic stress-strain data. This will more definitively establish the expected levels of scatter in the data just from experimental sources.
- A standard protocol should be established to report the chemical composition and microstructure of the test material as part of the reported data. The principles of quantitative metallography and the capabilities of automated imaging equipment should be fully utilized to obtain statistically meaningful results.
- Changes to the ASTM standards for CCG and CFCG testing must be made to include the results from the proposed study.
- Document a procedure for qualifying the test equipment used by participants for testing prior to commencing the tests.

**Funding:** The author wishes to acknowledge partial financial support from the Electric Power Research Institute (EPRI) under Independent Contractor Agreement 10010829 for the study.

**Data Availability Statement:** The data presented in this study are available on request from the corresponding author.

**Acknowledgments:** Useful discussions with Jonathan Parker, John Shingledecker, John Siefert, and Thomas Sambor, all from EPRI. Assistance from Santosh Narasimhachary of Siemens Technology is also gratefully acknowledged.

**Conflicts of Interest:** The authors declare no conflict of interest.

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
