# Peer review of "A Phenomenological Model for Creep and Creep-Fatigue Crack Growth Rate Behavior in Ferritic Steels"

_metals, doi:10.3390/met13101749_

Round 1

Reviewer 1 Report

The reviewer thinks that the manuscript is interesting and useful for engineers. But some parts are difficult for general readers to understand. The followings are comments.

 Introduction: The introduction is very short and does not include the background of this research. By introducing similar papers and stating the difference from your research, please emphasize the novelty of this manuscript.

 Eq.(2): Creep damage is evaluated based on Eq.(2) in which the distribution of creep cavities on grain boundaries is introduced. However, the results and discussions do not mention it. Is it possible to relate C* parameter obtained by experiment or literatures with the actual distribution of cavities? It seems that Eq.(2) and creep cavities are completely unrelated in the resutls.

 Lines 153-155: "there is a clear difference in the CCG rates between ex-service and new materials when samples are taken from the heat-affected zone (HAZ) regions." Describe its reason in relation to the cavity distribution.

Author Response

The introduction has been totally revised to address the comments of this reviewer but also pointed out by the other two reviewers.

The phenomenological model considers  idealized cavity sizes and distribution and can be used to primarily derive the form of temperature compensated crack growth rate, da/dt*, and a microstructural parameter related to average inter-cavity spacing. Extracting more details is not possible within an analytical model that uses idealized cavity distribution. There is no way to get any cavity distribution information from experimentally measured C* values. I do understand the reviewers comment but it is not possible to get microstructural changes due to cavity evolution into a deformation based parameter such as C*.

I have added a sentence to respond to this reviewer's last comment.

Reviewer 2 Report

I thank the author for such a great work and valuable article. Given my long history in this field, this paper was one of the best and most amazing manuscripts I have reviewed and read. Very valuable results have been reported and I believe that the author deserves to be encouraged (for example, in addition to free publishing, this article could be a candidate for the best article of the year in the journal). The manuscript has a proper structure and the written language is eloquent and understandable for the readers. The details are fully explained. However, some corrections are suggested before publishing.

1- Introduction section is very short and there is a lack of literature review, so, it is strongly suggested to extend this part. 

2- On page 5 line 148, this is only a question for my own information, the study  was done in weldment, and heat affected area. Were all samples of the same type of welding? Do not welding parameters affect creep life? Or, like the fatigue phenomenon, there is a master diagram that includes creep for all welding conditions. Please explain it. 

3- Regarding figure 2, the caption should change to "The basemetal".

4- On page 7 line 197, the reference format should change to [19].

5- Regarding Figure 5, the numbers of X-axis are not complete. 

6- It is better to change the place of section 5 and 6. It means that write summary and conclusion at first, then write recommendations for future research. 

Author Response

I thank the reviewer for the compliments and for a very thorough review. The responses match with the numbers used by the reviewer.

  1. The introduction has been totally revamped as per the suggestion of all three reviewers.
  2. Not enough information was available for distinguishing between weld types in the papers from the literature. This would be a good future study.
  3. done
  4. done
  5. Has been fixed
  6. Done

Reviewer 3 Report

This manuscript analyzes the effects of temperature and microstructural characteristics on the creep crack growth (CCG) and creep-fatigue crack growth (CFCG) behavior of ferritic steels. To rationalize these effects, a phenomenological model was proposed. This model was evaluated using extensive amounts of data gathered from the literature. The research is well designed and presented clearly. A quite short comparative analysis of existing publications is carried out, and some issues should be fixed. The methodological section of the manuscript is presented in sufficient detail. Extensive amounts of BM CCG and CFCG data for new and rejuvenated ex-service Grade 91 steels in the temperature range of 538 °C to 650 °C are reported from the literature. The tests were conducted on four separate heats of material. The developed model contains a characteristic microstructural parameter that can only be determined by quantitative microstructural evaluation of the test materials, typically not reported in CCG studies. It was found that the CCG in the BM of Grade 91 steels at temperatures of 538 °C, 593 °C, and some at 600 and at 650 °C follow creep-ductile tendencies implying higher CCG resistance than those under creep-brittle behavior observed in some heats tested at 600 °C and 625 °C. CCG constants for Grade 22 and Grade 91 steels representing the mean, upper bound (UB), and lower bound (LB) trends were also calculated and reported. For Grade 91, the constants were reported separately for creep-ductile and creep-brittle trends.

However, some shortcomings should be corrected to make the manuscript acceptable for publication in Metals.

(1) The Introduction section is too short. It should be expanded.

(2) Lines 37–38: The author should introduce the abbreviation WVSB.

(3) Line 43: It is expected that the author will explain shortly the meaning of the following parameters: “…da/dt versus crack tip parameters C* or Ct [5-8]” (in addition to the references provided). Line 65 has the same problem.

(4) The author should explain how the cavity radius at nucleation was determined.

(5) Line 151: The abbreviation CI should be introduced as follows: “…the 95% confidence interval (CI)…”.

(6) Line 191: The typo “in shown” should be deleted.

(7) Lines 197–198: Only the numbers in square brackets remain, and the authors' names are removed.

(8) More new References (2019–2023) should be added.

(9) References must be formatted in accordance with the journal's requirements.

Author Response

  1. Introduction is totally revamped
  2. done
  3. This is now done fully in the new introduction
  4. The nucleation cavity size is neglected as being negligible in comparison to the critical size in creep-ductile materials
  5. Done
  6. Done
  7. Done
  8. One new reference has been added but references are limited to ones that can be justified, whether from the author's research group or by others. This is not a very active area of research anymore so it is hard to find justifiable reasons to include more references from the past three years.
  9. I think the references are formatted as per the journal requirements.